# Assessment of Sex-Specific Associations between Athletic Identity and Nutrition Habits in Competitive Youth Athletes

**DOI:** 10.3390/nu16172826

**Published:** 2024-08-23

**Authors:** James J. McGinley, Nolan D. Hawkins, Taylor Morrison, Emily Stapleton, Emily Gale, Joseph Janosky, Henry B. Ellis, Sophia Ulman

**Affiliations:** 1Center for Excellence in Sports Medicine, Scottish Rite for Children, Frisco, TX 75034, USA; jamesmcginley@att.net (J.J.M.); taylor@taylored-nutrition.com (T.M.); henry.ellis@tsrh.org (H.B.E.); 2Department of Psychology, Scottish Rite for Children, Frisco, TX 75034, USA; emily.stapleton@tsrh.org (E.S.); emily.gale@tsrh.org (E.G.); 3Sports Medicine Institute, Hospital for Special Surgery, New York, NY 10021, USA; janoskyj@hss.edu; 4Department of Orthopaedic Surgery, University of Texas Southwestern, Dallas, TX 75390, USA

**Keywords:** psychology, sports, pediatrics, diet, adolescent, nutrition surveys, sports medicine

## Abstract

Given the psychological aspects of sports nutrition, understanding one’s athletic identity (AI) may improve targeted nutrition education. Therefore, the purpose of this study was to examine nutrition habits and AI among uninjured youth athletes. Athletic Identity Measurement Scale (AIMS) and custom Sports Nutrition Assessment for Consultation (SNAC) scores collected prospectively at local sporting events were retrospectively assessed via Mann–Whitney, Kruskal–Wallis, logistic regression, and ANCOVA tests (95% CI). Among 583 athletes (14.5 ± 2.1 years; 59.9% female), the total AIMS scores did not differ by sex (males 39.9 ± 7.2; females 39.3 ± 7.5; maximum 70). The Social Identity (*p* = 0.009) and Exclusivity (*p* = 0.001) subscores were higher in males, while the Negative Affectivity subscores were lower (*p* = 0.019). Females reported frequent associations between SNAC and AIMS, particularly Negative Affectivity, which was positively associated with stress fracture history (*p* = 0.001), meal-skipping (*p* = 0.026), and desiring nutrition knowledge (*p* = 0.017). Males receiving weight recommendations reported higher Negative Affectivity subscores (*p* = 0.003), and higher total AIMS scores were observed in males with fatigue history (*p* = 0.004) and a desire for nutrition knowledge (*p* = 0.012). Fatigue and stress fracture history predominated in high-AI males and females, respectively, suggesting that poor sports nutrition may present differently by sex. As suboptimal nutrition was frequently related to high Negative Affectivity subscores, these habits may increase following poor sports performance.

## 1. Introduction

Healthy nutrition habits are a foundational aspect of any effective sport training program, particularly in the developing youth athlete. However, competitive athletes often undergo physically demanding training regimens, resulting in the consumption of convenient snacks, sport drinks, and supplements with limited nutritional benefit [1]. Additionally, in certain sports, such as gymnastics and football, coaches often attempt to steer athletes towards an optimal body type which differs from their natural state [2]. While elite female athletes have traditionally been the focus of research on body image and eating disorders [3,4,5], any athlete may suffer negative effects from inadequate nutrition. These negative effects include greater incidence of food intolerance, changes in appetite after exercise, or meal-skipping for performance or esthetic reasons. Unfortunately, youth athletes may mistake unhealthy weight-loss diets as healthy performance-driven diets, therefore missing out on essential ingredients for exercise preparation and recovery such as carbohydrates, calcium, proteins, and fats [3,6]. Nutrition education programs have shown limited success in the current literature [7,8,9,10], and consequently, youth athletes’ knowledge of healthy and effective nutrition habits remains suboptimal [5].

Sports nutrition contains clear social and psychological components as youth athletes compare their habits and physique with their own ideal image and that of others [1,3]. As such, an understanding of athletic identity, or the degree to which an individual identifies with the athlete role in both an individual and social sense [11], may be a useful measure to better understand the risk for certain nutrition habits. The recent literature has reported that athletes with high athletic identity are more subject to compulsive over-training [12] and an exacerbated desire to alter their body composition [13,14]. The increased prevalence of eating disorders has also been related to higher athletic identity [15,16] or the associated exercise identity [17]. When athletes aim to cause such drastic changes to their physique, they may put themselves at risk for nutrition habits which can have both short- and long-term detriments to their health, particularly in those with less involved coaches. One method of adjusting body composition may be skipping meals, particularly breakfast [18], which can have negative effects on available energy for an athlete [19]. Though athletes may recognize this actual or perceived negative effect [20], the allure of weight loss or limits of time constriction may reduce meals eaten. However, to our knowledge, no studies have examined athletic identity regarding specific nutrition habits.

An evaluation of the relationship between athletic identity and eating habits is warranted due to the strong psychological component of nutrition and the potential deleterious effects of attempting to train for competitive athletics with improper or absent instruction. Disordered or unhealthy nutrition habits, while potentially developed during youth sport participation, may remain after sport retirement and lead to issues with body composition and eating habits with greater nutritional concerns than lifetime non-athletes [21]. A better understanding of which athletes are at risk for suboptimal nutrition habits through athletic identity may provide important information regarding targeted nutrition education which can be implemented by coaches or others who identify such athletes, hopefully decreasing the occurrence of nutrition-influenced injuries. Therefore, the purpose of this study was to examine the association between nutrition habits and athletic identity among a cohort of uninjured youth athletes. We hypothesized that those with higher athletic identity would display greater incidence of suboptimal nutrition habits, defined as affirmative responses to a custom nutrition questionnaire.

## 2. Materials and Methods

Following Institutional Review Board (IRB) approval, a cohort was retrospectively sampled of uninjured sport-specialized youth athletes enrolled in a prospectively collected Sports-Specific Assessment and Functional Evaluation (SAFE) program between 10 and 15 October 2023. Informed consent was obtained from all subjects involved in this study by verbal and written consent prior to any study procedures being undertaken, and patient rights were protected. Given that athletes were minors at the time of participation, all athletes were approached for consent with their parent or legally authorized representative available for prospective participation. Athletes who were active in sport and not currently injured were recruited for the program at youth athletic practices and camps. Surveys were completed electronically via tablets before, during, or shortly after the athletic event while still on-site and may have been completed by the child with help from the parent or by the child alone. Research assistants were available to provide instruction if requested from the athlete. Athletes were then included in this retrospective sub-analysis under the same IRB approval if they responded fully to both the Athletic Identity Measurement Scale (AIMS) and a custom institutional Sports Nutrition Assessment for Consultation (SNAC) survey, as some athletes chose not to complete all surveys. Duplicate instances of athletes who were recruited to participate in multiple events were excluded.

Athletes were administered the AIMS survey, during which they rated individual questions from 1 to 7, with higher scores indicating higher athletic identity [22]. The modified seven-item AIMS depicts three primary categories contributing to athletic identity, expressed as Social Identity (Questions 1–3), Exclusivity (Questions 4–5), and Negative Affectivity (Questions 6–7), and has been found to be valid and reliable for youth athlete use [22]. Social Identity refers to the degree to which the respondent identifies with being an athlete as well as the degree to which athletics influences their goals and social circle. Exclusivity refers to the importance of athletics in the respondent’s life, and Negative Affectivity refers to the effects of poor athletic performance and injuries which inhibit athletic participation [22].

The SNAC survey queried respondents about their food allergies, weight changes, history of fatigue/stress fracture, nutrition knowledge, and other key nutrition habits and experiences. The SNAC survey is detailed in Table 1. Demographic and sport characteristics were also collected, including years since specialization in a primary sport, competition level, training volume, primary sport, and sport at time of data collection.

The recent literature emphasizes the value of reporting AIMS item and subscale information [14]; therefore, the responses to the SNAC survey were compared to the individual AIMS questions, AIMS subscales, and total AIMS scores. Given the lack of an established threshold to dichotomize the AIMS scores into high and low athletic identity, the analysis results were reported as percent differences. Specifically, percent differences were calculated as the difference in AIMS scores between those who responded “Yes” and those who responded “No” divided by the mean AIMS score of summed “Yes” and “No” responses. Positive values indicated that the response “Yes” had a higher AIMS score and negative values indicated that the response “No” had a higher AIMS score. Preliminary analyses identified significant differences in AIMS individual and subscores by sex, consistent with the current literature [23]. Potential differences may also exist in nutrition habits [5] by sex, so all analyses were separated by sex. Percent differences compared by sex indicate the difference in AIMS scores between “Yes” responses in male and female athletes divided by the mean of all “Yes” responses. Given significant tests of normality, the analyses consisted of non-parametric Mann–Whitney U tests and Kruskal–Wallis tests, as appropriate. Binary logistic regression analysis was performed to calculate the odds ratio of each SNAC question receiving a “Yes” response from males in comparison to females. In order to consider the potential impacts of demographic and sport participation variables, Analysis of Covariance (ANCOVA) tests were run to identify relationships between total AIMS scores and each SNAC question. As primary sport closely resembled sport at injury in the current sample, it was not included among the covariates. Statistical analyses were performed using SPSS Statistics (IBM Corp., Armonk, NY, USA). A standard 95% confidence interval was used for all tests.

## 3. Results

A total of 583 athletes were included in the final cohort for analysis. Athletes were aged 14.5 ± 2.1 years (6.0–21.7 years) and the majority were female (59.9%), Not Hispanic/Latino (79.8%), and White (62.8%). Most athletes reported specializing in a primary sport (92.6%), and they specialized at an average of 9.6 ± 4.3 years old. Athletes played 10.7 ± 1.8 months per year with 4.3 ± 1.7 practices per week at a mostly select/club level (52.3%) of competition (Table 2).

For females, the mean total AIMS scores were 39.3 ± 7.5 (out of 70). The mean subscores included Social Identity of 18.8 ± 3.6 (out of 21), Exclusivity of 10.3 ± 2.7 (out of 14), and Negative Affectivity of 10.3 ± 3.0 (out of 14). The mean total AIMS scores for males were 39.9 ± 7.2, with subscores of 19.4 ± 3.0 for Social Identity, 10.8 ± 3.1 for Exclusivity, and 9.6 ± 3.2 for Negative Affectivity. All subscores were significantly different by sex. Specifically, males indicated greater Social Identity (*p* = 0.009) and Exclusivity (*p* = 0.001) while females indicated greater Negative Affectivity (*p* = 0.019). On average, females ate breakfast 5.2 ± 2.3 days per week, which did not differ by AIMS score (*p* = 0.289), while males ate breakfast 5.7 ± 2.0 days per week on average, which also did not differ by AIMS score (*p* = 0.378). However, breakfast frequency did significantly differ between males and females (*p* = 0.002).

Associations between SNAC and AIMS were found among female responses, particularly the Negative Affectivity subscale. Negative Affectivity scores were 7.5% higher in females who skipped meals (*p* = 0.026), 8.3% higher in females who desired to learn more about nutrition (*p* = 0.017), and 16.5% higher in females with a history of stress fractures (*p* = 0.001). Stress fracture history was also associated with higher scores on both individual Negative Affectivity questions: Question 6 (‘I feel bad about myself when I do poorly in sport’; 17.6%; *p* = 0.002) and Question 7 (‘I would be very depressed if I were injured and could not compete in sport’; 15.3%; *p* = 0.024). Moreover, females with a stress fracture history also reported higher Question 3 (‘Most of my friends are athletes’; 10.4%; *p* = 0.028) and total AIMS scores (8.5%; *p* = 0.023). Skipping meals was related to a 6.9% higher Question 6 score (‘I feel bad about myself when I do poorly in sport’; *p* = 0.034), and a desire to understand nutrition was related to a 9.4% higher Question 7 score (‘I would be very depressed if I were injured and could not compete in sport’; *p* = 0.038). Finally, having a food intolerance was associated with 3.5% lower Question 2 scores (‘I have many goals related to sport’; *p* = 0.031; Table 3).

In contrast to females, significantly higher Negative Affectivity subscores were only reported in males who had been recommended body composition changes (14.0%; *p* = 0.003). Athletes with body composition recommendations also reported higher Question 4 (‘Sport is the most important part of my life’; *p* = 0.048), Question 6 (‘I feel bad about myself when I do poorly in sport’; *p* < 0.001), and total AIMS scores (*p* = 0.002; Table 3). In fact, 26.5% of all males reported receiving a suggestion on their body composition in comparison to 11.2% of females, resulting in 2.9 times greater odds by sex (*p* < 0.001; Table 4). Males reported associations between the AIMS score and a desire to learn about nutrition, as well. These athletes reported 7.4% higher Question 1 (‘I consider myself an athlete’; *p* = 0.038), 13.9% higher Question 4 (‘Sport is the most important part of my life’; *p* = 0.042), 18.2% higher Question 6 (‘I feel bad about myself when I do poorly in sport’; *p* = 0.006), and 8.9% higher total AIMS scores (*p* = 0.012). Additionally, a history of fatigue was associated with 22.3% higher Question 5 (‘I spend more time thinking about sport than anything else’; *p* = 0.041), 34.9% higher Question 6 (‘I feel bad about myself when I do poorly in sport’; *p* = 0.003), and 15.1% higher total AIMS scores (*p* = 0.004; Table 3). Despite these associations, males were only 0.28 times as likely to report fatigue as females (3.0% vs. 10.0%; *p* = 0.002; Table 4). Finally, males who skipped meals were also those with higher Question 6 scores (‘I feel bad about myself when I do poorly in sport’; 17.3%; *p* = 0.002; Table 3). However, males again were only 0.61 times as likely to report skipping meals as females (17.5% vs. 25.8%; *p* = 0.020; Table 4). The raw AIMS scores are detailed in the Appendix A.

The ANCOVA results considering the effects of sex, ethnicity, race, sport at testing, and competition level found that the total AIMS scores remained significantly dependent on desiring nutrition knowledge (*p* = 0.002) and being given body suggestions (*p* = 0.014) in all athletes (Table 5).

## 4. Discussion

Nutrition is an integral factor in youth athlete training, and expanded studies on the specific components of nutrition are necessary for more targeted, effective screening and education [3]. The results of this study found athletic identity to be significantly associated with a number of nutrition-related measures. These associations were most frequent among the Negative Affectivity subscore, defined as an athlete’s response to poor performance or negative events. Higher athletic identity was repeatedly associated with more frequent meal-skipping, which has multiple interpretations. Athletes in sports that require a certain esthetic or body composition may be inclined to skip meals [2]. The challenge of juggling school, athletics, and personal health might also result in unintentional meal-skipping, which was not differentiated by the questionnaire. Though females are often targets of the unhealthy nutrition narrative for esthetic sports [3,4,5], and ANCOVA testing revealed significance after controlling for sex, suggestions to alter body composition were not found to significantly relate to AIMS scores in the female-only study sample. Nonetheless, males reported associations between AIMS score and suggestions to change their body composition, which may hint at a more purposeful reason for meal-skipping. It may also reflect a more overt method of body composition suggestions in males from coaches and trainers compared to more covert methods in females through social media and traditional athlete stereotypes. Regardless of these habits, athletes expressed a clear desire to learn more about nutrition for the purposes of recovery, and this interest only heightened with higher athletic identity after controlling for potential confounders. Despite previous work reporting limited nutrition knowledge [5] and difficulty in changing nutrition habits [7], these results are promising for improving youth athlete training programs.

Females with high athletic identity scores generally reported issues with a history of stress fractures, while males with high athletic identity scores reported a history of fatigue. However, these findings were not statistically significant when potential demographic variables were included in the model, though none of the additional covariates showed significance. Interestingly, though also not statistically significant, females with high athletic identity reported less fatigue history, while males with high athletic identity reported less stress fracture history. These associations were bolstered when compared directly by sex, suggesting that stress fractures are less tied to athletic identity in males and fatigue is less tied to athletic identity in females.

The difference in fatigue among males may be partially explained by the association with suggestions for body composition changes [13,14], as inconsistent eating patterns could disrupt the body’s energy supply [24]. Still, males in sports are often recommended to increase their food intake to supplement more intensive training programs, frequently centered around building strength. As such, it could be that male athletes are unable to sufficiently intake food to fuel these training programs and require additional expertise to prevent fatigue following unintentional calorie deficits. Associations with athletic identity were more varied in males than females, but Question 6 (‘I feel bad about myself when I do poorly in sport’; Negative Affectivity subscale) and total AIMS score were repeatedly related to nutrition responses. Though not related to Negative Affectivity, males who were suggested body composition changes had higher athletic identity scores than females who heard the same, which might indicate that this pattern is only present among the most motivated male athletes [14,23]. Males with high athletic identity may benefit from counseling on caloric intake, especially in the context of responding to negative feedback from a coach, a parent, or themselves. Among males who choose to limit their nutritious intake due to this negative feedback regarding sport, bone injuries may become more frequent in addition to injuries to other tissues [25].

Females have an increased risk of stress fractures compared to males [26], which, in conjunction with the current study’s findings, emphasize the possibility that stress fractures are a more common indicator of risky nutrition habits specifically in females. Females with a higher Negative Affectivity score had a higher propensity for meal-skipping, and females skipped breakfast more often regardless of athletic identity, which may partially explain this group’s higher stress fracture rates. As such, it is important for nutrition education for females to include more information regarding habits which might decrease the risk of stress fractures, such as increasing calcium and Vitamin D intake with consistent meals [27]. Prior research has suggested limited use of vitamin supplements, further emphasizing the importance of nutritious meals spanning across the day [1].

Overall, while different presentation patterns of suboptimal nutrition habits were observed among males and females, those with higher athletic identity appear to be at higher risk for fatigue or stress fracture. These findings were noticed among those with higher Negative Affectivity, which may indicate that athletes are most at risk for negative nutrition habits after poor performance or career-changing events. To minimize the effects of athletic performance on psychological status and subsequent physical health, supporters of youth athletes should aim to improve their knowledge of healthy nutrition. These educators may include coaches who screen their athletes or who coach competitive teams and are expected to have a high percentage of high-athletic-identity athletes. Educators may also include parents or healthcare professionals in the community who aim to improve the nutritional habits of athletes and promote safe methods of weight gain and weight loss, regular eating times, and a separation between diet and attempts to address poor sports performance. Specifically, youth athletes with high athletic identity should be educated on maintaining a baseline level of health despite high-stakes performance and busy training schedules which can derail steady nutrition [1]. If steady nutrition is interrupted, athletes may also gain weight instead of adopting unhealthy patterns of losing weight, which can also have sport-specific negative effects [28].

This study was able to examine associations between athletic identity and specific nutrition habits in a large cohort of youth athletes from multiple sports. However, athletes were often highly specialized from a young age, which could limit the generalizability of this study. The custom SNAC survey used in this study has not been validated, and it did not screen for unintentional meal-skipping or mental health concerns, disordered eating, and eating disorders. These factors may have an impact on athlete overuse injuries and fatigue [24]. Finally, athletes were recruited from a variety of sports with potentially different expectations for nutrition, but this diverse cohort offers a large-scale view of the youth athlete population regarding nutrition.

## 5. Conclusions

As the effects of disordered or unhealthy eating habits may last beyond retirement from youth sport, it is essential to promote good nutrition habits among youth athletes. This study reported that athletes who tie their identity to sport more closely—especially those who suffer greater psychological effects from negative performance—tend to skip more meals and hear more comments regarding their body composition, but they are more interested in learning about nutrition. Males and females also report higher incidence of fatigue and stress fracture, respectively, with higher athletic identity scores. These results suggest that males and females with a strong sense of athletic identity both suffer from poor nutrition habits despite a traditional focus on females, yet the specific consequences of these habits differ by sex. Given that athletes in this study reported a strong willingness to learn about nutrition, nutrition knowledge must continue to expand in importance among youth athlete training programs in a sex-specific manner. This expansion is especially important among competitive athletes whose performance and inclusion in sport may influence psychological and physical well-being, such as those with a high athletic identity.

## Figures and Tables

**Table 1 nutrients-16-02826-t001:** Sports Nutrition Assessment for Consultation (SNAC) survey.

Variable Name	Question
Food intolerance	Do you have any food allergies/intolerances or avoid any food groups?
Appetite changes	Have you experienced any recent changes in appetite?
Skipping meals	Do you regularly skip at least one meal a day?
Desiring nutrition knowledge	Do you wish you better understood nutrition for your training and/or recovery?
Weight changes	Have you experienced any recent intentional or unintentional changes in your weight?
Body suggestions	Are you trying or has someone recommended that you change your body composition or weight?
Fracture history	Do you have a history of stress fractures?
Fatigue history	Do you struggle with dizziness or fatigue during games, practices or with exercise?
During the past 7 days, on how many days did you eat breakfast?	

All questions prompted a “Yes” or “No” response, with the exception of the last question regarding breakfast which was answered from 0 to 7.

**Table 2 nutrients-16-02826-t002:** Athlete characteristics.

Descriptives		*n*	%
Sex	Female	349	59.9
	Male	234	40.1
Race	Native American/Alaskan Native	7	1.2
	Asian	39	6.7
	Black/African American	166	28.5
	Native Hawaiian or Pacific Islander	5	0.9
	White	366	62.8
	Other	30	5.1
	Prefer Not to Answer	22	3.8
Ethnicity	Hispanic/Latino	74	12.7
	Not Hispanic/Latino	465	79.8
	Prefer Not to Answer	36	6.2
Sport at Testing	Baseball	12	2.1
	Basketball	163	28.0
	Cross Country	16	2.7
	Gymnastics	26	4.5
	Soccer	189	32.4
	Volleyball	166	28.5
	Other	11	1.9
Primary Sport	Basketball	135	23.2
	Cross Country	11	1.9
	Gymnastics	24	4.1
	Soccer	186	31.9
	Softball/baseball	15	2.6
	Track/field	7	1.2
	Volleyball	154	26.4
	Other	8	1.4
Competition Level	National Elite	98	16.8
	College or Professional	22	3.8
	Select/club (Travel or Non-Travel)	305	52.3
	High school	169	29.0
	Middle school	41	7.0
	Recreational	12	2.1
	Not reported	243	41.7

Athletes could select multiple races. Sport responses totaling *n* < 7 were tabulated as “Other”.

**Table 3 nutrients-16-02826-t003:** AIMS and SNAC Scores by Sex.

	AIMS	Male (*n* = 234)		Female (*n* = 349)		Sex Comparison	
		ΔAIMS	*p*-Value	ΔAIMS	*p*-Value	ΔAIMS	*p*-Value
Food Intolerance		Yes: 26 (11.1%)		Yes: 52 (14.9%)			
	Total	−3.3%	0.206	0.4%	0.480	1.3%	0.606
	SI	1.4%	0.801	−2.0%	0.475	−6.1%	0.122
	EX	−7.5%	0.120	−0.1%	0.583	1.5%	0.987
	NA	−8.0%	0.318	2.4%	0.706	15.6%	0.082
Appetite Changes		Yes: 24 (10.3%)		Yes: 25 (7.2%)			
	Total	0.7%	0.852	−2.4%	0.959	−4.3%	0.787
	SI	0.0%	0.495	−5.9%	0.324	−8.7%	0.303
	EX	−3.9%	0.808	−4.6%	0.351	−6.1%	0.324
	NA	7.4%	0.283	6.0%	0.230	5.2%	0.498
Skipping Meals		Yes: 41 (17.5%)		Yes: 90 (25.8%)			
	Total	1.7%	0.308	1.9%	0.291	−1.4%	0.603
	SI	−1.4%	0.491	−2.4%	0.148	−3.7%	0.172
	EX	0.1%	0.741	4.1%	0.429	−2.5%	0.178
	NA	9.9%	0.080	7.5%	0.026 *	3.8%	0.520
Desiring Nutrition Knowledge		Yes: 180 (76.9%)		Yes: 262 (75.1%)			
	Total	8.9%	0.012	4.2%	0.062	−2.3%	0.392
	SI	6.6%	0.157	1.2%	0.581	−4.3%	0.002
	EX	11.6%	0.074	5.8%	0.089	−6.6%	0.001
	NA	10.2%	0.103	8.3%	0.017 *	6.0%	0.026
Weight Changes		Yes: 28 (12.0%)		Yes (28, 8.0%)			
	Total	1.9%	0.291	−3.4%	0.907	−6.2%	0.511
	SI	0.9%	0.302	−7.6%	0.088	−11.2%	0.014
	EX	2.0%	0.582	−3.3%	0.769	−10.2%	0.098
	NA	3.7%	0.344	4.3%	0.186	6.9%	0.464
Body Suggestions		Yes: 62 (26.5%)		Yes: 39 (11.2%)			
	Total	7.3%	0.002	1.6%	0.713	−5.2%	0.109
	SI	4.0%	0.279	0.4%	0.595	−5.7%	0.028
	EX	7.2%	0.060	1.2%	0.961	−9.5%	0.020
	NA	14.0%	0.003	4.3%	0.197	0.2%	0.799
Fracture History		Yes: 12 (5.1%)		Yes: 32 (9.2%)			
	Total	−2.7%	0.779	8.5%	0.023	8.7%	0.244
	SI	1.1%	0.230	5.3%	0.227	0.5%	0.264
	EX	−2.5%	0.931	6.4%	0.312	2.7%	0.739
	NA	−10.7%	0.356	16.5%	0.001	30.7%	0.005
Fatigue History		Yes: 7 (3.0%)		Yes: 35 (10.0%)			
	Total	15.1%	0.004	−2.9%	0.844	−17.6%	0.005
	SI	6.4%	0.129	−4.5%	0.252	−13.2%	0.030
	EX	17.8%	0.080	−7.6%	0.145	−28.2%	0.003
	NA	29.7%	0.104	4.8%	0.302	−14.8%	0.085

Δ = percent difference. For the Male and Female columns, percent difference was calculated as the difference between “Yes” and “No” answers divided by the mean of all responses. For the Sex Comparison, the percent difference was calculated as the difference between male “Yes” and female “Yes” responses divided by the mean of all “Yes” responses. Statistical significance notated by an asterisk (*). AIMS = Athletic Identity Measurement Scale. SI = Social Identity. EX = Exclusivity. NA = Negative Affectivity. Positive percentage indicates that a “Yes” answer was related to higher AIMS or females had higher AIMS.

**Table 4 nutrients-16-02826-t004:** Sex differences in SNAC responses.

SNAC Question	*n*	Odds Ratio (95% CI)	*p*-Value
Food intolerance	583	0.71 (0.43–1.17)	0.189
Appetite changes	583	1.48 (0.82–2.67)	0.189
Skipping meals	583	0.61 (0.40–0.92)	0.020 *
Desiring nutrition knowledge	583	1.11 (0.75–1.64)	0.609
Weight changes	583	1.56 (0.90–2.71)	0.115
Body suggestions	583	2.87 (1.85–4.49)	<0.001 *
Fracture history	583	0.54 (0.26–1.04)	0.074
Fatigue history	583	0.28 (0.11–0.60)	0.002 *

Binary logistic regression results. Odds ratios indicate odds of males reporting a “Yes” response in comparison to females. Statistical significance notated by an asterisk (*). CI = confidence interval.

**Table 5 nutrients-16-02826-t005:** ANCOVA results for AIMS scores by SNAC question.

SNAC Question	*df*	SS	MS	F	*p*-Value
Food intolerance	1	0.430	0.430	0.008	0.927
Corrected Model	6	264.2	44.04	0.862	0.523
Appetite changes	1	94.52	94.52	1.861	0.173
Corrected Model	6	358.3	59.72	1.176	0.319
Skipping meals	1	141.9	141.9	2.803	0.095
Corrected Model	6	405.8	67.63	1.335	0.241
Desiring nutrition knowledge	1	758.1	758.1	15.54	<0.001 *
Corrected Model	6	1021.9	170.3	3.491	0.002 *
Weight changes	1	84.35	84.35	1.660	0.199
Corrected Model	6	348.2	58.03	1.142	0.338
Body suggestions	1	305.7	305.7	6.095	0.014 *
Corrected Model	6	569.5	94.92	1.893	0.081
Fracture history	1	123.0	123.0	2.426	0.120
Corrected Model	6	386.8	64.47	1.271	0.270
Fatigue history	1	1.484	1.484	0.029	0.865
Corrected Model	6	265.3	44.22	0.866	0.520

Between-subjects effects for total AIMS scores. Covariates included sex, ethnicity, race (White vs. Non-White), sport at time of testing, and highest competition level (high school and below vs. club and above). Statistical significance notated by an asterisk (*). *df* = degrees of freedom. SS = Sum of Squares. MS = Mean Square.

## Data Availability

The data are contained within the article or Appendix A.

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
