# Peer review of "Assessment of Sex-Specific Associations between Athletic Identity and Nutrition Habits in Competitive Youth Athletes"

_nutrients, 2024, doi:10.3390/nu16172826_

Round 1

Reviewer 1 Report

Comments and Suggestions for Authors

1.The logic of “…those with higher athletic identity would display greater incidence of suboptimal nutrition habits” was not clearly described enough.  

2.Since in the Discussion section nearly each question was described, I suggest that all seven questions of AIMS should also show in Table 3 as three primary categories and total AIMS did.

3.For AIMS, its reliability and validity should be reported in the manuscript. Especially, the reliability using the samples in the present research should be reported.

4. In the Abstract, “chi-square” was reported to use, but in the Method section or other sections, it was not mentioned. However, in line 110, “Kruskal-Wallis test” was mentioned.

5. Table 2 presented the background data. However, these may influence the results of Table 3 and Table 4 if they were not considered in statistical analyses. Logistic regression tests can accommodate these covariates.

6. If the authors will consider accommodate those background variables into statistical analyses, please also present different tests (ANOVA) for AIMS by background variables.

7. The normal distribution of AIMS variables were seriously violated? In fact, ANOVA and t-test have robustness, using ANCOVA can accommodate those covariates in Table 2 to replace Kruskal-Wallis test and Mann-Whitney U tests.

Author Response

1. Summary

Thank you very much for taking the time to review this manuscript. We have included detailed responses below, and the corresponding revisions/corrections are highlighted in the re-submitted files.

2. Questions for General Evaluation

Reviewer’s Evaluation

Response and Revisions

Does the introduction provide sufficient background and include all relevant references?

Can be improved

See below.

Is the research design appropriate?

Can be improved

Are the methods adequately described?

Can be improved

Are the results clearly presented?

Can be improved

Are the conclusions supported by the results?

Yes

3. Point-by-point response to Comments and Suggestions for Authors

 Comments 1: The logic of “…those with higher athletic identity would display greater incidence of suboptimal nutrition habits” was not clearly described enough.  

Response 1: Thank you for this comment. To address this confusion in our hypothesis, the phrase “defined as affirmative responses to a custom nutrition questionnaire” was added. We believe this more accurately describes the hypothesis of the study in a statistical sense as we now expect higher athletic identity to relate to more “yes” responses on the SNAC.

Comments 2: Since in the Discussion section nearly each question was described, I suggest that all seven questions of AIMS should also show in Table 3 as three primary categories and total AIMS did.

 Response 2: Thank you for your comment. We agree that question-specific data is important in the interpretation of the results as it is frequently discussed in the Discussion paragraph. However, due to the large volume of data that this creates, we felt it would be best included in the Supplementary Material so that readers are not repeatedly exposed to the same results in both paragraph and tabular form. For that reason, we have chosen to keep the question-specific results in the Supplementary Material for this revised version.

Comments 3: For AIMS, its reliability and validity should be reported in the manuscript. Especially, the reliability using the samples in the present research should be reported.

Response 3: Thank you for your comment. We did not assess reliability and validity using the current sample, but prior research has confirmed that the AIMS is valid and reliable. This has been added to the Methods section as “and has been found valid and reliable for youth athlete use [18].”

Comments 4: In the Abstract, “chi-square” was reported to use, but in the Method section or other sections, it was not mentioned. However, in line 110, “Kruskal-Wallis test” was mentioned.

Response 4: Thank you for noticing this! We have corrected the abstract and updated it to also include the new ANCOVA which was done per your suggestion.

Comments 5: Table 2 presented the background data. However, these may influence the results of Table 3 and Table 4 if they were not considered in statistical analyses. Logistic regression tests can accommodate these covariates.

Response 5: Thank you for your comment. While we attempted to correct for any potential changes by sex in the original manuscript, we acknowledge other factors may have an impact. The current method of presenting results has been retained as we feel it offers an easily understood and applicable way for readers to identify how each SNAC question changes athletic identity; however, we have also chosen to add an ANCOVA test which assesses the impact of each covariate in Table 2 (except for primary sport, which was too similar to sport at testing but with less data) on the relationship between SNAC questions and total AIMS score. We believe that by including this ANCOVA it will bolster our results in Table 3 without overwhelming the reader with additional data on the AIMS subscores. Please see the new Table 5.

Comments 6: If the authors will consider accommodate those background variables into statistical analyses, please also present different tests (ANOVA) for AIMS by background variables.

Response 6: Thank you for this comment. We have made the appropriate changes to introduce an ANCOVA into the statistics.

Comments 7: The normal distribution of AIMS variables were seriously violated? In fact, ANOVA and t-test have robustness, using ANCOVA can accommodate those covariates in Table 2 to replace Kruskal-Wallis test and Mann-Whitney U tests.

Response 7: Thank you for your comment. As mentioned above, we have chosen to introduce an ANCOVA test for total AIMS score and SNAC questions to improve the interpretation of our results while keeping the non-parametric tests which offer easy visualization of the relationship between the SNAC and AIMS.

Reviewer 2 Report

Comments and Suggestions for Authors

This paper depeens the relationship between athletic Identity and nutrition habits in competitive youth athletes, with particular regard to sex peculiarities.

It is interesting and in line with this Journal aims, but some aspects have to be clarified and corrected.

Introduction seems too short. It should be expanded in order to explain why the authors started this research and which literature gap they are searching to fill. 

Materials and methods should be globally revised. There is an IRB approval, this is important but not sufficient. How was the informed consent collected for the athletes involved? It is mandatory also in retrospective study, and moreover it is mandatory for minors, as your sample subjects are in the majority minors (avarage age was 14.5 ± 2.1). Please calrify it.

Another concern is the recruitment process. You said "uninjured sport-specialized youth athletes enrolled in a prospectively colected functional evaluation program between October 10th, 2022 – October 15th, 2023". Which program? Why shall we consider them as a homogeneous sample, since they all practice different sports? The survey distribution is not clear and should be explained (which mean did you use? and how did you instructed the participants to complete it?).

You defined the enrolled subjects as athletes: what the meaning of this word you referred to? As you know, there are many difference between athletes, amateurs, sportpeople, agonists and not-agonists and so on. According to what you reported, they are not all at a competitive level.

How did you relate fatigue and fractures with previous injuries?

Results are clearly presented.

In my opinion, the discussion is too short and it should be integrated better explaining how your findings could help future strategies for preventing injury in professional athletes. To do that, I suggest the following references:

-Farì, G., Fischetti, F., Zonno, A., Marra, F., Maglie, A., Bianchi, F. P., Messina, G., Ranieri, M., & Megna, M. (2021). Musculoskeletal Pain in Gymnasts: A Retrospective Analysis on a Cohort of Professional Athletes. International journal of environmental research and public health18(10), 5460. https://doi.org/10.3390/ijerph18105460

-Farì, G., Quarta, F., Longo, S. C., Masiero, L., Ricci, V., Coraci, D., Caforio, L., Megna, M., Ranieri, M., Varrassi, G., & Bernetti, A. (2024). How does classification score affect falls in wheelchair basketball? A video-based cross-sectional study on the Italian national team during the European Para Championships 2023. Physical therapy in sport : official journal of the Association of Chartered Physiotherapists in Sports Medicine67, 77–82. https://doi.org/10.1016/j.ptsp.2024.03.006

Conclusions are clear and in line with results.

Best regards

Author Response

General comment: This paper depeens the relationship between athletic Identity and nutrition habits in competitive youth athletes, with particular regard to sex peculiarities.

It is interesting and in line with this Journal aims, but some aspects have to be clarified and corrected.

1. Summary

Thank you very much for taking the time to review this manuscript. We have included detailed responses below, and the corresponding revisions/corrections are highlighted in the re-submitted files.

2. Questions for General Evaluation

Reviewer’s Evaluation

Response and Revisions

Does the introduction provide sufficient background and include all relevant references?

Can be improved

Please see below.

Is the research design appropriate?

Must be improved

Are the methods adequately described?

Must be improved

Are the results clearly presented?

Yes

Are the conclusions supported by the results?

Can be improved

 3. Point-by-point response to Comments and Suggestions for Authors

Comments 1: Introduction seems too short. It should be expanded in order to explain why the authors started this research and which literature gap they are searching to fill.

Response 1: Thank you for this comment. We have added additional information regarding our inspiration for this research to the Introduction. We would like to point out that our literature gap is the following: “However, to our knowledge, no studies have examined athletic identity regarding specific nutrition habits.” We know that nutrition can have a large impact on an athlete’s mental status, and we also know that an athlete’s sense of their athletic self can cause them to take drastic action, such as severe over- or under-fueling, in order to improve their performance. With this in mind, we think it is critical to obtain a better understanding of the relationship between nutritional habits and athletic identity. Please see the highlighted sections of the Introduction for exact additions.

Comments 2: Materials and methods should be globally revised. There is an IRB approval, this is important but not sufficient. How was the informed consent collected for the athletes involved? It is mandatory also in retrospective study, and moreover it is mandatory for minors, as your sample subjects are in the majority minors (avarage age was 14.5 ± 2.1). Please calrify it.

 Response 2: Thank you for your concern! We have added a number of clarifications to the Methods. All athletes provided informed consent with their parent or guardian available, and this consent was collected in verbal and written form by a research assistant. This retrospective sub-analysis did not require a separate IRB approval as the prospective study provided for future analyses, but this has also been clarified in the manuscript. Please see the first paragraph of the Methods for exact verbiage.

Comments 3: Another concern is the recruitment process. You said "uninjured sport-specialized youth athletes enrolled in a prospectively colected functional evaluation program between October 10th, 2022 – October 15th, 2023". Which program? Why shall we consider them as a homogeneous sample, since they all practice different sports? The survey distribution is not clear and should be explained (which mean did you use? and how did you instructed the participants to complete it?).

Response 3: Thank you for this comment. The name of the program was initially excluded to ensure anonymity of the authors. However, we agree that this is an important method of recruitment, so we have added the name of this program with a black highlight. The purpose of this program is to evaluate athletic youth in terms of movement, nutrition, psychosocial variables, and activity level to develop normative data in healthy athletes. As such, athletes are identified through community sports events which partner with our institution and allow us to approach all of their members for participation. Though different sports are included in the dataset, this sample comprises a large set of competitive athletes to create a picture of the generalized youth athlete.

In terms of specifics regarding survey distribution, the following clarification has been added to the Methods: “Surveys were completed electronically via tablets before, during, or shortly after the athletic event while still on-site and may have been completed by the child with help from the parent or the child alone. Research assistants were available to provide instruction if requested from the athlete.” Potential participants were recruited upon entrance to the athletic event, and they completed these surveys at the same time either before, during, or after their event.

Comments 4: You defined the enrolled subjects as athletes: what the meaning of this word you referred to? As you know, there are many difference between athletes, amateurs, sportpeople, agonists and not-agonists and so on. According to what you reported, they are not all at a competitive level.

Response 4: Thank you for your comment. In prior literature, we have found that “athlete” is a broad term which can apply to both competitive and non-competitive level as long as the individual competes in some sort of athletic events. As all participants were recruited at local athletic events, we feel this best categorizes our sample to provide readers with the most accurate description of who was tested, given that as you said, athletes and non-athletes may have very different attitudes and habits. We acknowledge that any naming convention will have limitations, but we believe the current convention to be best suited for the current manuscript.

Comments 5: How did you relate fatigue and fractures with previous injuries?

Response 5: Unfortunately, data was not available regarding prior injuries given that this was a retrospective review of prospectively-collected data. In future work, we would absolutely be interested in validating this question to the occurrence of injuries or exhaustion events.

Comments 6: In my opinion, the discussion is too short and it should be integrated better explaining how your findings could help future strategies for preventing injury in professional athletes. To do that, I suggest the following references:

-Farì, G., Fischetti, F., Zonno, A., Marra, F., Maglie, A., Bianchi, F. P., Messina, G., Ranieri, M., & Megna, M. (2021). Musculoskeletal Pain in Gymnasts: A Retrospective Analysis on a Cohort of Professional Athletes. International journal of environmental research and public health18(10), 5460. https://doi.org/10.3390/ijerph18105460

-Farì, G., Quarta, F., Longo, S. C., Masiero, L., Ricci, V., Coraci, D., Caforio, L., Megna, M., Ranieri, M., Varrassi, G., & Bernetti, A. (2024). How does classification score affect falls in wheelchair basketball? A video-based cross-sectional study on the Italian national team during the European Para Championships 2023. Physical therapy in sport : official journal of the Association of Chartered Physiotherapists in Sports Medicine67, 77–82. https://doi.org/10.1016/j.ptsp.2024.03.006

Response 6: Thank you for this comment. Additional discussion has been added to discuss potential takeaways from the article, though we want to remain focused on the youth athletic sphere rather than professional athletics given most participants fall in the middle school, high school, and club levels of competition. We found the first article pertinent to our discussion as higher BMI, perhaps due to poor nutrition habits, may have negative effects on an athlete’s sport performance in a sport-specific way. The exact changes to the Discussion may be found in the highlighted text of the revised manuscript, but nearly all paragraphs have been expanded.

Round 2

Reviewer 1 Report

Comments and Suggestions for Authors

OK

Reviewer 2 Report

Comments and Suggestions for Authors

Thank you for the efforts to improve the quality of your paper according to my suggestions. I don't think there are any further corrections that need to be made to the manuscript at this point.